# A Simple Differentiation Protocol for Generation of Induced Pluripotent Stem Cell-Derived Basal Forebrain-Like Cholinergic Neurons for Alzheimer’s Disease and Frontotemporal Dementia Disease Modeling

**DOI:** 10.3390/cells9092018

**Published:** 2020-09-02

**Authors:** Sonia Sanz Muñoz, Martin Engel, Rachelle Balez, Dzung Do-Ha, Mauricio Castro Cabral-da-Silva, Damian Hernández, Tracey Berg, Jennifer A. Fifita, Natalie Grima, Shu Yang, Ian P. Blair, Garth Nicholson, Anthony L. Cook, Alex W. Hewitt, Alice Pébay, Lezanne Ooi

**Affiliations:** 1Illawarra Health and Medical Research Institute, Wollongong, NSW 2522, Australia; ssm886@uowmail.edu.au (S.S.M.); mengel@uow.edu.au (M.E.); rb478@uowmail.edu.au (R.B.); pddh859@uowmail.edu.au (D.D.-H.); mcastro@uow.edu.au (M.C.C.-d.-S.); tberg@uow.edu.au (T.B.); 2School of Chemistry and Molecular Bioscience, University of Wollongong, Wollongong, NSW 2522, Australia; 3Department of Anatomy & Neuroscience, University of Melbourne, Parkville, VIC 3010, Australia; Damian.Hernandez@unimelb.edu.au (D.H.); apebay@unimelb.edu.au (A.P.); 4Department of Surgery, University of Melbourne, Parkville, VIC 3010, Australia; Alex.Hewitt@utas.edu.au; 5Centre for Eye Research Australia, Royal Victoria Eye and Ear Hospital, Melbourne, VIC 3002, Australia; 6Centre for Motor Neuron Disease Research, Department of Biomedical Sciences, Faculty of Medicine and Health Sciences, Macquarie University, Sydney, NSW 2190, Australia; jennifer.fifita@mq.edu.au (J.A.F.); natalie.grima@mq.edu.au (N.G.); shu.yang@mq.edu.au (S.Y.); ian.blair@mq.edu.au (I.P.B.); 7Concord Repatriation General Hospital, University of Sydney ANZAC Research Institute, Sydney, NSW 2006, Australia; garth.nicholson@sydney.edu.au; 8Wicking Dementia Research and Education Centre, University of Tasmania, Hobart, TAS 7001, Australia; anthony.cook@utas.edu.au; 9School of Medicine, Menzies Institute for Medical Research, University of Tasmania, Hobart, TAS 7000, Australia

**Keywords:** induced pluripotent stem cells, disease modelling, neuronal differentiation, cholinergic neurons, Alzheimer’s disease, frontotemporal dementia

## Abstract

The study of neurodegenerative diseases using pluripotent stem cells requires new methods to assess neurodevelopment and neurodegeneration of specific neuronal subtypes. The cholinergic system, characterized by its use of the neurotransmitter acetylcholine, is one of the first to degenerate in Alzheimer’s disease and is also affected in frontotemporal dementia. We developed a differentiation protocol to generate basal forebrain-like cholinergic neurons (BFCNs) from induced pluripotent stem cells (iPSCs) aided by the use of small molecule inhibitors and growth factors. Ten iPSC lines were successfully differentiated into BFCNs using this protocol. The neuronal cultures were characterised through RNA and protein expression, and functional analysis of neurons was confirmed by whole-cell patch clamp. We have developed a reliable protocol using only small molecule inhibitors and growth factors, while avoiding transfection or cell sorting methods, to achieve a BFCN culture that expresses the characteristic markers of cholinergic neurons.

## 1. Introduction

The use of induced pluripotent stem cells (iPSCs) has opened the possibility to generate relevant cell types that contain the genetic background of each donor, allowing the study of neurodegenerative diseases in a reductionist system. The cholinergic system is affected in dementias, including Alzheimer’s disease (AD) and frontotemporal dementia (FTD) [1,2]. AD is a progressive and irreversible neurodegenerative disease characterized by cognitive impairment and memory loss. Currently, the drugs that are prescribed to AD patients target symptoms and show limited efficacy [3]. AD patients undergo neurodegeneration throughout the brain, however basal forebrain-like cholinergic neurons (BFCNs), part of the cholinergic system and characterized by the use of the neurotransmitter acetylcholine (ACh), are the first type of neurons to degenerate [1,2]. Basal forebrain atrophy also occurs in FTD, with degeneration of the cholinergic system thought to be causative in learning, and memory in FTD patients [4,5]. The potential selective vulnerability of the BFCNs, however has to date been difficult to study in human neurons.

AD can be divided into two forms: familial AD (FAD), also known as early-onset AD, and sporadic AD (SAD), also known as late-onset AD. FAD makes up less than 5% of all cases, and it is characterised by mutations in the amyloid precursor protein (*APP*) or presenilin (*PSEN*) genes, both of which are involved in amyloid β (Aβ) processing [6]. In contrast, SAD is the most common form, where the possession of the ε4 allele of apolipoprotein E (*APOE*) gene is the major genetic risk factor [7,8,9], but aging, environmental and lifestyle factors also influence the risk.

To ensure successful differentiation of BFCNs in vitro, a combination of growth factors and small molecule inhibitors mimicking in vivo BFCN development are important, and very few protocols have been developed, as reviewed in [10]. The published protocols to generate BFCNs used a combination of adherent and floating cultures, also involving transfection and cell sorting based purification, which dramatically affects yields [11,12,13]. Differences in BFCN purity and efficiency (based on choline acetyl transferase (ChAT) positive cells) have been identified in the published protocols, which is a major challenge to maintaining experimental consistency when multiple cell lines are compared.

The aim of the present study was to generate and characterise a robust protocol to differentiate a pure culture of BFCNs from iPSCs and validate it by gene expression and functional characterisation. The model described provides a relevant cell type to study AD pathways and for drug discovery.

## 2. Materials and Methods

### 2.1. iPSC Lines and iPSC Maintenance

The details of the iPSC lines used in this study are described in Table 1. 

Human feeder-free iPSCs were cultured on Matrigel (Corning, #354277) coated 60 mm diameter tissue culture dishes fed with TeSR-E8 Basal Medium (Stemcell Technologies, #5990) and kept in a humidified incubator at 37 °C, 5% CO_2_ and 3% O_2_, mimicking the physiological oxygen levels in the brain. After manual removal of spontaneously differentiating cells under the microscope, the medium was changed every day. Colonies were passaged every 5–6 days using 1 U/mL dispase (Stemcell Technologies, #7913) for 5 min at 37 °C. Once the border of the colonies started to detach from the surface, cells were washes off twice with Dubelcco’s modified Eagle medium/Nutrient Mixture F-12 (DMEM/F12). DMEM/F12 was added a third time, colonies were mechanically detached with a cell scraper, transferred into a conical centrifuge tube and centrifuged for 5 min at 300× *g*. The supernatant was removed, and colonies were gently resuspended in 1 mL TeSR-E8 before being plated down at a 1:5 ratio. The iPSC cultures were tested routinely for mycoplasma and karyotyping of all the iPSC lines was performed within 20 passages before the start of the differentiation.

### 2.2. Generation of Forebrain Cholinergic Neurons

Following iPSC culture, TeSR-E8 medium was changed to Neural induction (Ni) medium (0.4% (*v/v*) B27 supplement (Life Technologies (Carlsbad, CA, U.S.A.), #17504044), 1% (*v/v*) N2 supplement (Life Technologies, #17502048), 1% (*v*/*v*) Non-essential amino acids (Life Technologies, #11140050) and 1% (*v/v*) GlutaMAX (Life Technologies, #35050061) in DMEM/F12) supplemented with 0.1 μM LDN193189 (Focus Bioscience, #12071A), a Bone morphogenic protein (BMP) inhibitor 2 days before passaging. Medium was changed if necessary, depending on the density of the cultures, avoiding acidic conditions. 

On the passaging day, cells were incubated with 1 U/mL dispase until colonies fully detached from the plate. Colonies were collected and rinsed twice with DMEM/F12 to remove the enzyme, allowing them to settle by gravity for 2 min at 22 °C. The supernatant was removed, and colonies were resuspended in Ni media supplemented with 0.1 μM LDN193189 and 10 μM of the Transforming Growth Factor (TGF)-β inhibitor SB431542 (Focus Bioscience, #10431), pipetting gently up and down to break down the colonies. Colonies were transferred into a low adherent non-tissue culture plate to form floating embryoid bodies (EBs). Full medium change with Ni media containing 0.1 μM LDN193189 and 10 μM SB431542 was performed every second day. 

EBs were collected on day 5 and plated down into Matrigel-coated tissue culture plates with Ni media supplemented with 10 ng/mL Fibroblast Growth Factor (FGF)-2 (Global Stem, #GDR-2001) to form neural rosettes. A partial volume media change with Ni medium supplemented with 10 ng/mL FGF-2 was performed after 48 h and with 10 ng/mL FGF-2 and 50 ng/mL SHH (Stemcell Technologies, #78075) after 96 h. Neural rosette formations were visible by day 7. 

Once neural rosettes were visible on day 7, medium was removed, and cells were incubated with 1 U/mL dispase until the colony edges loosened. Dispase was removed and cells were carefully rinsed 3 times with DMEM/F12 to ensure the removal of dispase and cell debris. Neural expansion (Ne) medium (2% (*v/v*) B27 supplement, 1% (*v/v*) N2 supplement, 1% (*v/v*) Non-essential amino acids and 1% (*v*/*v*) GlutaMAX in DMEM/F12) supplemented with 100 ng/mL Sonic Hedgehog (SHH) was added and identified neural rosettes were detached by aspiration using a P1000 pipette. Floating neural rosettes were transferred into a low adherent non-tissue culture plate to form floating neurospheres. Half volume media changes were performed every second day using Ne medium supplemented with 100 ng/mL SHH for the first 6 days; 100 ng/mL SHH and 100 ng/mL FGF-8 (Stemcell Technologies, #78128) from day 6 until day 12; and 100 ng/mL SHH, 100 ng/mL FGF-8 and 10 ng/mL BMP9 (Peprotech, #120-07) on day 12. 

On day 15 of neurosphere stage, neurospheres were washed twice with DMEM/F12 prior dissociation with StemPro Accutase (Life Technologies, #A1110501) for 20 min at 37 °C. Enzymatic dissociation was stopped by dilution with DMEM/F12 and cells were recovered by centrifugation at 300× *g* for 5 min. Cells were counted and 125,000 cells/well were plated on Matrigel and 0.1 mg/mL collagen I (Thermo Fisher Scientific, #A1048301) -coated 24-well plate format. 

Neuronal maturation (Nm) medium (1% (*v/v*) B27 supplement and 1% (*v/v*) GlutaMAX in Brainphys (Stemcell Technologies, #5790)) supplemented with 100 ng/mL of SHH, 100 ng/mL of FGF-8, 10 ng/mL of BMP9 and 100 ng/mL of Nerve Growth Factor (NGF) (Peprotech, #450-01) was used on days 1 and 3 of cholinergic maturation. On day 5, a half volume media change using Nm media supplemented with 5 ng/mL of BDNF (Miltenyi Biotech, #130-093-811) and 100 ng/mL of NGF in addition to 1 μM 5-Fluoro-2′-deoxyuridine (5FdU, Sigma-Aldrich, #F0503) to stop the proliferation of the dividing cells was used. A full media change was performed on day 7 using Nm medium supplemented with 5 ng/mL Brain Derived Neurotrophic Factor (BDNF) and 100 ng/mL NGF. Partial volume media changes using Nm medium with 5 ng/mL BDNF and 10 ng/mL NGF were performed every second day for a total of 4 weeks to allow neuronal maturation. 

### 2.3. Cell Imaging and Immunostaining

At week 4 of neuronal maturation, Nm medium was removed, neurons were rinsed with phosphate-buffered saline (PBS) and fixed by incubation with 4% (*w/v*) paraformaldehyde (PFA) at room temperature. After 10 min, PFA was removed and cells were rinsed three times with PBS, permeabilised with 0.5% (*v/v*) Triton X-100 in PBS for 15 min and rinsed three times with PBS. Blocking was performed with 5% (*v/v*) goat serum in PBS and 0.3 M glycine for 1 h at 22 °C. Primary antibodies were used at the specific dilution in 5% (*v/v*) goat serum in PBS and incubated at 4 °C for 16 h. On the following day, cells were rinsed three to five times with PBS and incubated with the appropriate secondary antibody made up in PBS for 1 h at 22 °C. Cells were washed three to five times with PBS before incubation with Hoechst 33,342 (Thermo Fisher Scientific, #62249, 1 μg/mL) or Reddot 2 (Biotium, #40061-T, 1:200) for 10 or 20 min respectively for nuclear staining. Coverslips were mounted on glass slides using ProLong Gold antifade reagent (Life Technologies, #P10144). Confocal microscopy was performed using Leica TCS SP8 (Leica Microsystems, Germany) microscopy images were analysed using Leica Application Suite - Advanced Fluorescence (LAS-AF) software. The antibodies used for immunocytochemistry are listed in Table 2. 

### 2.4. Nanostring

A custom CodeSet was designed to be used with the PlexSet Nanostring technology (Nanostring) to analyse the genes in Appendix A. The mRNA from iPSC and BFCN samples was harvested in TriSURE (Bioline, #38032) and extracted following the manufacturer’s instructions. The mRNA concentration of the samples was measured using a Qubit 3.0 Fluorometer (Thermo Fisher Scientific). To determine the optimal amount of mRNA per sample needed to not saturate the Nanostring, a titration run was performed and analysed using the nSolver System (Nanostring). A total of 100 ng mRNA per sample were run in the nCounter SPRINT (Nanostring). To normalize the samples, a reference sample compiled of all the samples was run alongside each Probe Set, and a total of 10 housekeepers *(AARS*, *ASB7*, *CCDC127*, *CNOT10*, *EID2*, *MTO1*, *RABEP2*, *SUPT7L*, *TADA2B*, *ZNF324B*) were selected based on the literature and Nanostring recommendations. Samples generated were analysed for each gene independently showing the number of molecules counted by the nCounter SPRINT System after normalization by housekeepers using the nSolver System.

### 2.5. Whole Cell Patch Clamp

Whole-cell patch-clamp was performed on the BFCNs after 4 weeks of maturation to assess the neuronal functionality. All experiments were performed at 22–24 °C in artificial cerebral spinal fluid (aCSF, 135 mM NaCl, 5 mM KCl, 2 mM CaCl2, 1 mM MgCl2, 10 mM HEPES and 10 mM D-glucose; adjusted to pH 7.4 (with NaOH) and ~305 mOsm) and using intracellular solution (150 mM KCl, 2 mM MgCl2, 10 mM HEPES, 4 mM Mg-ATP, 0.3 mM Na-GTP, 10 mM Na2PCr and 1 mM EGTA; adjusted to pH 7.4 (with KOH) and ~295 mOsm). Cells were visualised with an inverted microscope (Leica DM RB) and chosen based on neuronal morphology, which included large cell bodies and long extensions. Patching electrodes were pulled by a Sutter P97 from a 1.5 mm OD × 0.86 mm ID borosilicate glass (with filament) (Harvard Apparatus) to a resistance of 5–7 MΩ. Electrophysiological signals were recorded at 10 Hz using Digidata 1550 and amplified using a MultiClamp 700B amplifier (Axon Instruments, Molecular Devices Electrophysiology). The acquired data were analysed using the pCLAMP10 software. Whole-cell patch clamp configuration was achieved after establishing a GΩ seal (>1 GΩ). Cells with a series resistance (Rs) below 25 MΩ were used for further analysis. Membrane Capacitance (CM), Rs and resting membrane potential (RMP) were determined using the Membrane Test and pClamp10 analysis software. To determine firing properties of differentiated neurons, direct current was injected to hold the cell at −60 mV. In current clamp mode, step currents of 25 pA were injected form −50 pA until the cell fired or 500 pA. Synaptic events were recorded in voltage clamp mode by injecting direct current, holding the cell at −60 mV and performing a chart recording over 5 min. 

### 2.6. Data and Statistical Analysis

Ten iPSC lines described in Table 1 were used for the optimization and characterization of the BFCN protocol. These lines include healthy controls, SAD, FAD and ALS/FTD lines in order to represent more variability in the population rather than several differentiations within each cell line. 

The statistical analysis and graphs were performed using Prism 8 (GraphPad Prism 8). Each set of genes for iPSCs and BFCNs were checked for normality and therefore analysed using two-tailed *t*-test for those with parametric distribution and Kolmogorov-Smirnov for those with non-parametric distribution. The use of each test on the data is explained in the figure legends. 

## 3. Results

The generation of BFCNs from iPSCs included adherent and suspension stages, with the addition of small molecule inhibitors and growth factors to mimic biochemical signals received during development. An overview of each stage of the protocol to generate BFCNs is represented in Figure 1. Incubation of iPSCs for 48 h prior to generation of embryoid bodies with Ni media supplemented with the BMP inhibitor LDN193189, was followed by the addition of LDN193189 and the TGF-β inhibitor SB431542 in EB stage, in order to promote the ectoderm fate of the iPSCs. This could be observed by the formation of neural rosette-like structures inside the EBs (Figure 1C) and the generation of neural rosettes 5 days after plating the EBs onto Matrigel-coated plates (Figure 1D). Addition of FGF−2 in this stage stimulated neural progenitor expansion, and the formation of neural rosettes ensured the selection of the cells mimicking the neural tube formation by visual screening (Figure 1D). The neural rosettes were manually selected and kept in suspension to generate floating neurospheres. This structure allowed the expansion of the neural progenitors (Figure 1E) and, in the presence of SHH induces ventralization of the neural tube, the region from where BFCNs develop [19]. In the late stage of neurosphere expansion, the addition of FGF−8 mimicked the telencephalon development, and BMP9 induced the cholinergic phenotype of the progenitors [20]. As the last stage of the protocol, neurospheres were dissociated and plated as single cells, allowing the neural progenitors to mature as BFCNs (Figure 1F). Specific growth factors such as BDNF, which stimulates cholinergic differentiation, and NGF, which promotes neuronal maturation, arborization and survival, were required to promote the BFCN fate and increase ChAT activity [21,22,23]. The addition of 5-Fluoro−2′deoxyuridine (5FdU), which inhibits the proliferative cells by stopping DNA biosynthesis [24,25], helps the culture to focus on the furthest developed neurons and therefore can be used to generate highly pure neuronal cultures.

Immunocytochemistry for general neuronal and specific cholinergic markers was performed at 4 weeks of maturation in BFCN cultures. The cholinergic marker ChAT and the neuronal marker β-III-tubulin were identified in the BFCN cultures. Visual counting showed that 87% of the β-III-tubulin positive cells were also ChAT positive (Figure 2A–C). Immunocytochemistry against the NGF receptor p75NTR was also performed, with only a few cells identified as p75NTR+ve in the cultures and colocalised with β-III-tubulin (Figure 2D–F). The mRNA expression of the BFCN samples was analysed by Nanostring and compared to iPSC samples after adjusting to the expression of 10 housekeeping genes. The housekeepers were chosen to cover different ranges of expression (highly expressed, such as *AARS* or *CCDC127*, and lower expressed, such as *RABEP2* or *ZNF324B*). The expression of each housekeeper gene is shown in Appendix A and there was no significant difference in the overall housekeeper expression between iPSCs and BFCNs (Appendix A).

To characterise the BFCN cultures, sets of genes for different stages of development were analysed, and their expression compared to iPSC cultures. As expected, the pluripotency markers *NANOG* and *POU5F1* were highly expressed in iPSCs, whereas no expression was detected in BFCN cultures (Figure 3A,B). The RNA molecule counts for the pluripotency markers were in the range of 10,000-35,000 for all lines (Appendix A). The neuronal progenitor markers achaete-scute homolog 1 (*ASCL1*), distal-less homeobox 1 and 2 (*DLX1* and *DLX2*), empty-spiracles homeobox 1 (*EMX1*), paired box 6 (*PAX6*) and SRY-box 1 (*SOX1*) were analysed (Figure 3C–H). *EMX1* did not show a significant difference in expression between BFCN and iPSC cultures (Figure 3F). Assessing the RNA molecule count data showed that in all three SAD lines, but no other lines, an increase in *EMX1* RNA was detected in the BFCNs, compared to iPSCs (Appendix A). However, all other neuronal progenitor genes were upregulated in BFCN cultures compared to iPSCs: *ASCL1* was upregulated by 17.1 ± 5.2 fold (*p* < 0.001, D = 1), *DLX1* was upregulated by 54 ± 23 fold (*p* = 0.003, D = 0.8), *DLX2* was upregulated by 34.6 ± 10.7 fold (*p* < 0.001, D = 1), *PAX6* was upregulated by 16.6 ± 4.3 fold (*p* = 0.003, D = 0.8), and *SOX1* was upregulated by 35.2 ± 11 fold (*p* < 0.001, D = 1). The specific BFCN progenitor markers analysed were *FOXG1*, *ISL1*, *LHX8*, and *NKX2−1* (Figure 3I–L). While *FOXG1*, *ISL1* and *NKX2−1* were upregulated by 73.6 ± 15.9 fold (*p* < 0.001, D = 0.9), 53.3 ± 17.2 fold (*p* < 0.001, D = 0.9) and 7.2 ± 3.7 fold (*p* = 0.03, D = 0.8), respectively; *LHX8* did not show a significant difference between cultures. Of the BFCN progenitor markers, the most consistently upregulated gene across the 10 lines was *FOXG1* (Appendix A).

General neuronal markers such as solute carrier family 6 member 3 (*SLC6A3*, commonly known as dopamine transporter 1 (DAT1)), glutamate decarboxylase 2 (*GAD2*), glutamate ionotropic receptor AMPA type subunit 1 and 2 (*GRIA1* and *GRIA2*), glutamate ionotropic receptor NMDA type subunit 1 (*GRIN1*), microtubule associated protein 2 (*MAP2*), disc large MAGUK scaffold protein 4 (*DLG4*, commonly known as postsynaptic density protein 95 (PSD95)), synapsin I (*SYN1*), tyrosine hydroxylase (*TH*) and β-III-tubulin (*TUBB3*) were also analysed in iPSC and BFCN cultures (Figure 4A–J).

The glutamate decarboxylase *GAD2* and the glutamate receptors *GRIA1*, *GRIA2* and *GRIN1* were upregulated in BFCN when compared to iPSCs by 99.3 ± 34.5 fold (*p* = 0.01, t = 2.9), 76.3 ± 11.2 fold (*p* < 0.001, t = 6.8), 285.7 ± 23 fold (*p* < 0.001, D = 1) and 387.6 ± 59.1 fold (*p* < 0.001, D = 1), respectively. The RNA molecule counts across all 10 lines showed a consistent upregulation of the genes encoding the AMPA and NMDA receptor subunits, *GRIA1, GRIA2, GRIN1* (5000–20,000 counts)*,* as well as the neuronal cytoskeletal and synaptic protein genes, *MAP2* (100,000-200,000 counts)*, TUBB3, DLG4* (PSD95) and *SYN1* (10,000-40,000 counts; Appendix A). An upregulation of *TH* by 194.6 ± 33.5-fold (*p* < 0.001, t = 5.8) was also found in BFCN cultures, whereas the dopaminergic associated transporter DAT1 did not differ when compared to iPSCs. *MAP2* and *TUBB3*, which had also been identified in the BFCN cultures by immunostaining (Figure 2B,E and Figure 5F), were upregulated by 59.3 ± 3 fold (*p* < 0.001, t = 19.91) and 10.6 ± 0.9 fold (*p* < 0.001, t = 11.45), respectively. Moreover, the pre-synaptic marker *SYN1* and the post-synaptic marker PSD95 were upregulated by 11.7 ± 1.1-fold (*p* < 0.001, t = 10.73) and 7.05 ± 0.9 (*p* < 0.001, D = 1), respectively. The cholinergic cell markers *ACHE*, *CHAT*, solute carrier family 5 member 7, (*SLC5A7*, commonly known as CHT1), nerve growth factor receptor (*NGFR*), neurotrophic receptor tyrosine kinase 1 (*NTRK1*, also known as TRKA) and solute carrier family 18 member 3 (*SLC18A3*, commonly known as VACHT) were analysed (Figure 4K–P). The cholinergic neuron marker *ACHE* and *CHT1* were upregulated in BFCN cultures when compared to iPSCs by 50.2 ± 8.2 (*p* < 0.001, t = 6.2) and 19 ± 9.1 (*p* < 0.001, D = 1), respectively. However, the expression of *CHAT* and *NGFR* showed no significant difference between samples in the iPSC and BFCN cultures. The RNA molecule counts for *ACHE* were more consistently upregulated across all 10 lines (5000–20,000 counts), compared to *CHAT, SLC5A7* and *SLC18A3* (Appendix A). Moreover, from the specific NGF receptors analysed, *NTRK1* was upregulated by 1.9 ± 0.7 (*p* = 0.003, D = 0.8), while *NGFR* did not show a significant difference when BFCNs were compared to iPSCs. However, iPSCs showed high RNA molecule expression in *NGFR* (Appendix A) and this gene was not further upregulated following differentiation into BFCNs. These data together confirm the cholinergic fate of the BFCN cultures.

To further characterize the purity of the BFCN cultures, the astrocyte markers aldolase (*ALDOC)*, solute carrier family 1 member 3 (*SLC1A3*, commonly known as glial high affinity glutamate transporter 1 (EAAT1)), nuclear factor I A (*NFIA*) and S100B calcium binding protein B (*S100B*) were analysed (Figure 5A–D). *ALDOC* and *NFIA* did not show a significant difference when BFCN cultures were compared to iPSCs, whereas *S100B* was upregulated by 3.6 ± 1.4 (*p* < 0.001, D = 0.9) and *EAAT1* was downregulated by 0.6 ± 0.3 (*p* < 0.001, D = 0.9). RNA molecule counts for the astrocyte marker genes were low in the majority of lines; in many but not all lines, the levels of the astrocyte marker genes were lower in BFCNs than in iPSCs (Appendix A). BFCN cultures were also positive for the dendritic marker microtubule-associated protein 2 (*MAP2*), but no astrocytes were detected after staining against the activated astrocyte marker Glial fibrillary acid protein (*GFAP*) (Figure 5E–G), indicative of a highly pure neuronal culture. 

Whilst the expression of protein markers was useful to validate cholinergic neuron gene expression, neurons are ultimately defined by their electrical properties. Consequently, we analysed the electrophysiological properties of the iPSC-derived BFCNs. The functional properties of the BFCN cultures were analysed by whole-cell patch clamp at week 4 of BFCN maturation. A total of 45 cells across six BFCN samples were patch-clamped, showing an average resting membrane potential of −32 ± 2 mV and cell capacitance of 21 ± 2 pF (Figure 6A,B). Of the neurons analysed, 29% were able to fire action potentials when the membrane potential was held at −60 mV (Figure 6C), and 36% of the cells showed aborted spikes below 0 mV. Synaptic events were also recorded in BFCN cultures (Figure 6D).

## 4. Discussion

The generation of BFCNs from iPSCs can provide an important model to understand neurodevelopment or the neurobiology of AD and FTD, as this neuronal population is drastically reduced in AD and FTD patients [2]. This work establishes a reliable protocol to differentiate BFCNs cultures as shown here by differentiation of multiple cell lines by mimicking the in vivo signals required in development to generate BFCNs. The protocol relies on the use of small molecule inhibitors and growth factors avoiding transfection or cell sorting as other published protocols [11,12,13] to achieve a BFCN culture that expresses the characteristic markers of cholinergic neurons and negligible contamination with GFAP positive astrocytes (Figure 5).

The use of the nCounter SPRINT System (Nanostring) allows the simultaneous analysis of a large number of genes from the same pool of cells, ensuring a reliable characterization of the BFCNs. Here were show the gene expression of markers from different stages of neuronal differentiation as a relative expression of the BFCN cultures normalised to iPSCs (Figure 3 and Figure 4). Moreover, the relative amount of gene expression for each individual sample after housekeeper gene normalization is provided (Appendix A).

Neural developmental genes showed higher expression levels in the BFCNs compared to the iPSCs (Figure 3), except *EMX1*, a transcription factor related with the development of the cerebral cortex and not the forebrain region [26,27], that showed no significantly different expression in BFCNs when compared to iPSCs. *ASCL1*, a key gene expressed in neural progenitor cells that promotes the cell cycle exit [28,29], and *DLX1* and *DLX2*, regulated by *ASCL1* in the forebrain [30] showed increased expression in BFCN cultures when compared to iPSCs. *SOX1* and *PAX6*, another two important transcription factors for the development of neuroectodermal commitment, and specific cholinergic progenitor markers such as *FOXG1* and *ISL1*, required for forebrain fate [31,32] and *NKX2−1*, that determines the medial ganglion eminence fate, showed higher expression in BFCN cultures when compared to iPSCs. However, *LHX8* did not show a significant difference in BFCN and iPSC cultures, even if three of the BFCN samples showed higher expression (Figure 3K); but it has been shown that some subsets of cholinergic neurons can be generated in the absence of Lhx8 [33]. Overall, the expression of neuronal developmental markers and specific BFCN developmental markers indicates the expected fate towards BFCN neurons of the cells in culture.

From all the general neuronal markers analysed (Figure 4A–J), only *DAT1*, a dopamine transporter, did not show a significant increase in BFCN cultures when compared to iPSCs. All neuronal markers, including *TUBB3* and *TH* expression, the dendritic marker *MAP2*, the pre- and post-synaptic markers *SYN1* and PSD95 and the glutamate receptors *GAD2*, *GRIA1*, *GRIA2* and *GRIN1* showed a significant increase in BFCN cultures when compared to iPSCs. However, to ensure that specific BFCNs were generated in culture, the specific BFCN markers related to the synthesis and transport of ACh were analysed (Figure 4K–P). *ACHE*, the choline transporter CHT1 and the high affinity NGF receptor TRKA were upregulated; whereas *CHAT*, the low affinity NGF receptor *NGFR*, and VACHT did not show a significant difference between BFCN and iPSC cultures. However, previous studies have shown that *NGFR* plays a role in pluripotency [34], justifying its presence in iPSC cultures. Moreover, double immunocytochemistry of ChAT and β-III-tubulin showed an efficiency of 87% ChAT positive neurons (Figure 2A–C) indicating the cholinergic fate of the cultures.

Astrocytic markers were analysed to determine the purity of the BFCN cultures. From the four astrocytic markers analysed, *ALDOC* and *NFIA* did not show a significant difference, whereas *S100B* was upregulated and *EAAT1* was downregulated in the BFCN culture when compared to the iPSCs (Figure 5A–D). Given that immunocytochemistry could not identify GFAP positive staining in the cultures (Figure 5E–G), and mRNA counts of astrocyte markers were low, this indicates that the current protocol generates a highly pure culture of BFCNs.

This protocol was developed to allow for analysis of neurons, without contamination from astrocytes, as shown by the lack of GFAP staining and other astrocyte markers. Analysis of the functional properties of the BFCNs was determined by whole cell patch clamp, with neurons showing action potentials and synaptic events when the membrane potential was held at −60 mV (Figure 6C,D). Whereas recent publications have shown a resting membrane potential values of −40 mV in differentiated BFCNs [13,35], in this study the average resting membrane potential was −30 mV which, taken together with the high expression of some developmental markers, show the immaturity of the neurons. An increase in the maturation time over the current 4 weeks of the protocol or the presence of other cell types, such as astrocytes, would improve maturation of neurons in culture [36]. Analysis of co-cultures can be the subject of future research.

AD has mostly been studied in the late stages to understand the pathology of the disease. However, aging in a dish remains a limitation (as reviewed in [37]), even if some studies have already shown different approaches that may be used to model late-onset diseases. Manipulation of telomerases [38] or treatment with progerin [39] have induced the aging phenotype in iPSC-derived neurons modelling Parkinson’s disease (PD). Like PD, AD is a neurodegenerative disease whereby deposition of amyloid plaques or neurofibrillary tangles starts decades before the cognitive symptoms appear. Post mortem tissue from patients with the Lewy body variant of AD (dementia with Lewy bodies) also show a loss of cholinergic neurons and reduced ChAT activity very early in the disease course [40]. The iPSC-derived BFCNs can therefore help us understand the underlying pathological mechanisms that occur in the early stages of a range of neurodegenerative diseases.

## 5. Conclusions

The generation and characterization of a robust protocol to generate BFCNs from iPSCs has been described in this study. The BFCNs generated open a new way to unravel the specific neuropathology of BFCNs in AD and FTD.

## Figures and Tables

**Figure 1 cells-09-02018-f001:**
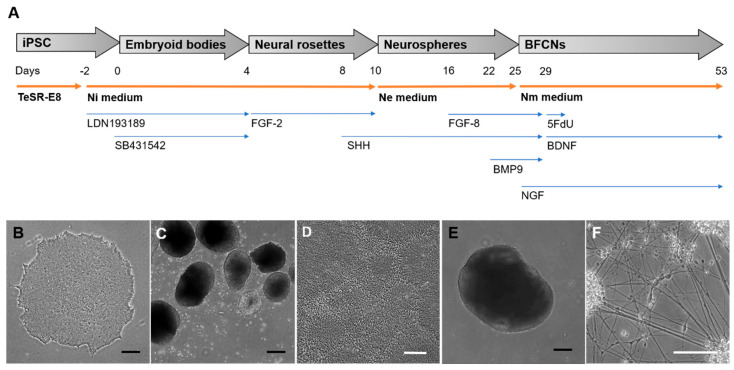
Differentiation of BFCNs from iPSCs. The iPSCs were differentiated into BFCNs by the combination of small molecule inhibitors and specific growth factors mimicking the developmental signals. (**A**) Schematic diagram showing the different stages of the differentiation in grey arrows over time. Orange arrows showing the cell culture media used in each stage and blue arrows showing the small molecule inhibitors or growth factors used at each specific time points. (**B**–**F**) Representative phase contrast pictures of the last day at each stage: (**B**) iPSC, (**C**) embryoid bodies, (**D**) neural rosettes, (**E**) neurospheres and (**F**) BFCNs. Black scale bars = 200 μm and white scale bars = 100 μm.

**Figure 2 cells-09-02018-f002:**
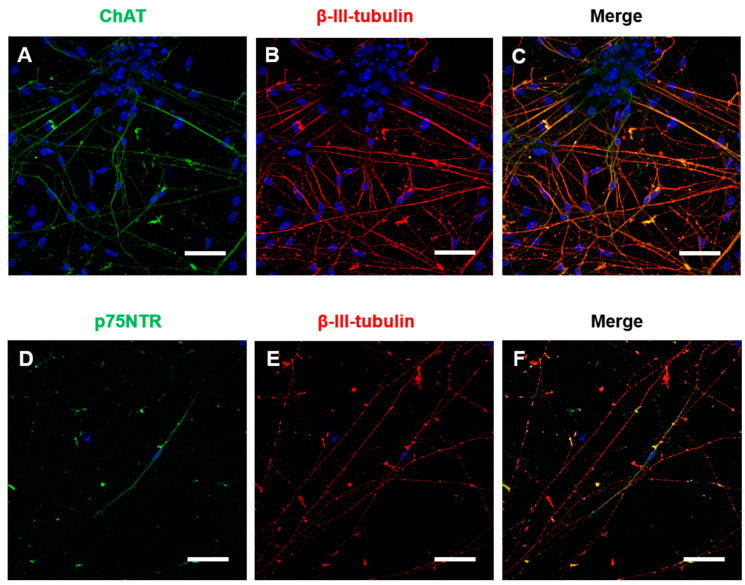
BFCNs were positive for neuronal markers. BFCN cultures were matured for 4 weeks and immunocytochemistry for general neuronal markers and specific cholinergic neuronal markers was performed. (**A**–**F**) Representative confocal pictures of the BFCN cultures at week 4 of BFCN maturation with the nuclear marker Hoechst 33,342 in blue. (**A**) Cholinergic marker ChAT in green. (**B**) Neuronal marker β-III-tubulin in red. (**C**) Merged picture of A and B. (**D**) Cholinergic marker p75NTR in green. (**E**) Neuronal marker β-III-tubulin in red. (**F**) Merged picture of D and E. Scale bars = 50 μm.

**Figure 3 cells-09-02018-f003:**
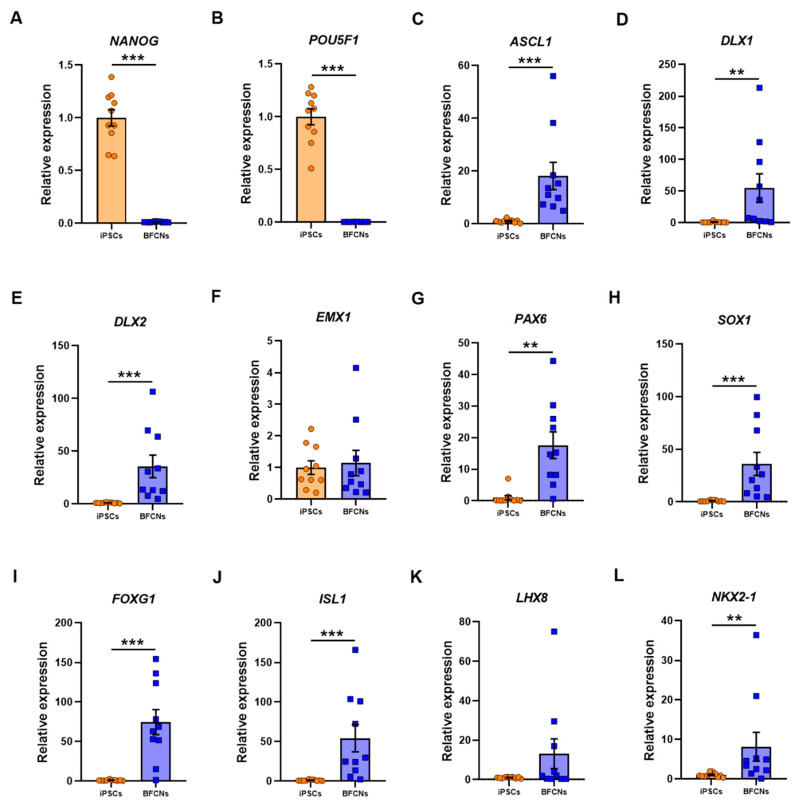
Expression of pluripotency, neuronal progenitors and specific cholinergic progenitor markers in iPSC and BFCN cultures. The iPSC and BFCN samples were analysed by nCounter (Nanostring) and, after normalisation of total amount of RNA molecules to the housekeeper genes, results are shown as fold change of BFCN cultures over iPSCs. Analysis of the pluripotency markers (**A**) *NANOG* and (**B**) *POU5F1*; the neuronal progenitor markers (**C**) *ASCL1*, (**D**) *DLX1*, (**E**) *DLX2*, (**F**) *EMX1*, (**G**) *PAX6*; and (**H**) *SOX1*; and the specific cholinergic progenitor makers (**I**) *FOXG1*, (**J**) *ISL1*, (**K**) *LHX8* and (**L**) *NKX2−1*. Data are derived from 10 iPSC lines and 10 BFCN differentiated samples from 1 independent differentiation for each line. Individual data are shown. Histogram bars represent mean values and error bars represent S.E.M. Graphs show ** *p* ≤ 0.01, *** *p* ≤ 0.001, by Kolmogorov-Smirnov test except in A and B that two-tailed *t*-test was used for parametric distribution.

**Figure 4 cells-09-02018-f004:**
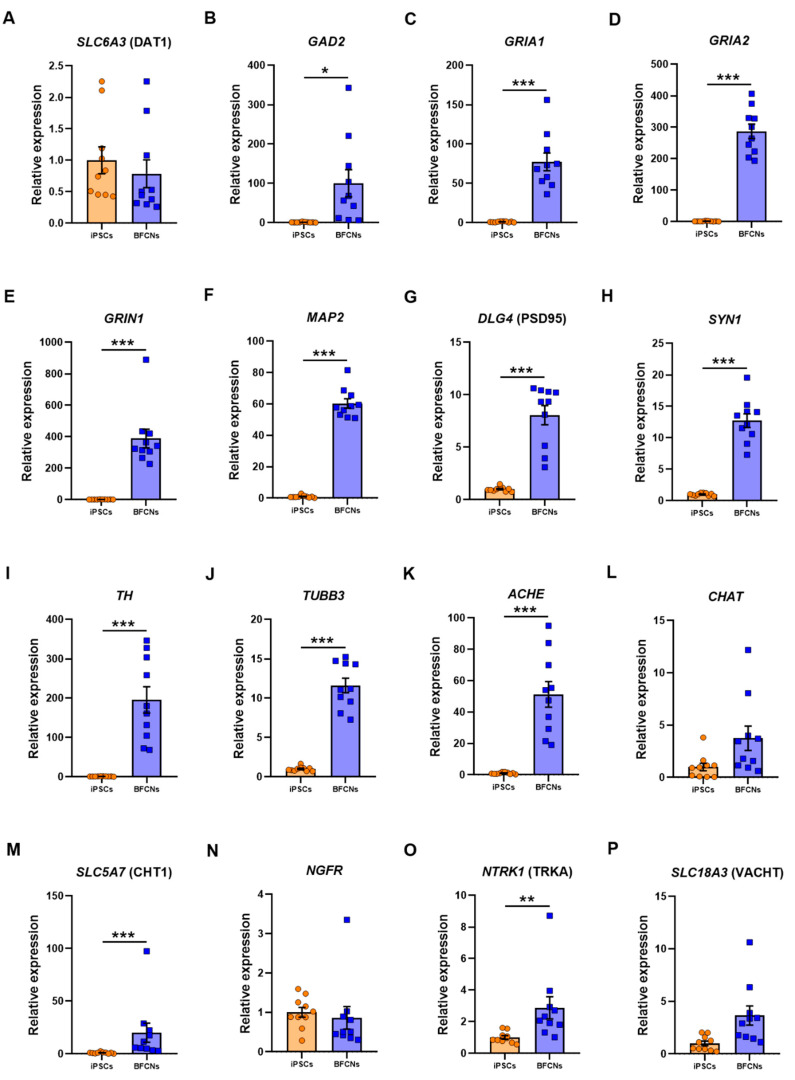
Expression of neuronal and specific BFCN markers in iPSC and BFCN cultures. The iPSC and BFCN samples were analysed by nCounter (Nanostring) and, after normalisation of total amount of RNA molecules to the housekeeper genes, results are shown as fold change of BFCN cultures over iPSCs. Analysis of the neuronal markers (**A**) *SLC6A3* (DAT1), (**B**) *GAD2*, (**C**) *GRIA1*, (**D**) *GRIA2*, (**E**) *GRIN1*, (**F**) *MAP2*, (**G**) *DLG4* (PSD95), (**H**) *SYN1*, (**I**) *TH* and (**J**) *TUBB3*; and the specific BFCN markers (**K**) *ACHE*, (**L**) *CHAT*, (**M**) *SLC5A7* (CHT1), (**N**) *NGFR*, (**O**) *NTRK1* (TRKA) and (**P**) *SLC18A3* (VACHT). Data are derived from 10 iPSC lines and 10 BFCN differentiated samples from 1 independent differentiation for each line. Individual data are shown. Histogram bars represent mean values and error bars represent S.E.M. Graphs show * *p* ≤ 0.05, ** *p* ≤ 0.01, *** *p* ≤ 0.001, by two-tailed *t*-test except in A, D, E, G, L, M, N, O and P that Kolmogorov-Smirnov test was used for non-parametric distribution.

**Figure 5 cells-09-02018-f005:**
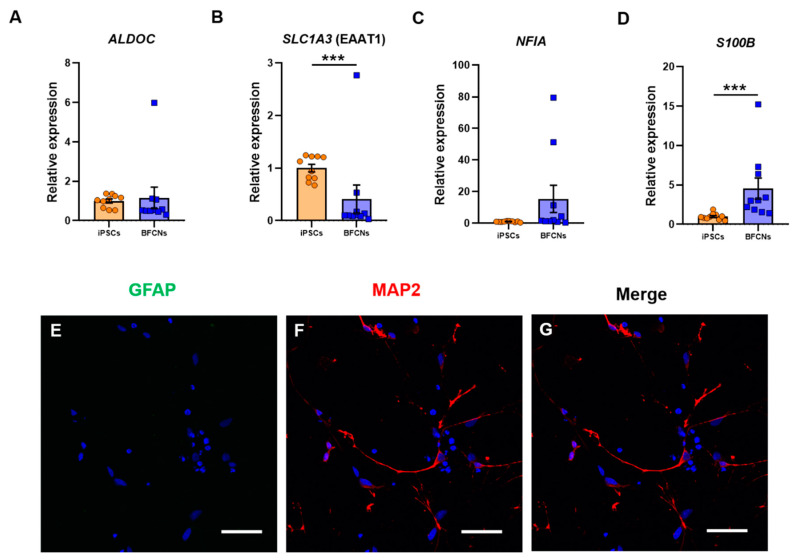
Expression of astrocytic markers. The iPSC and BFCN samples were analysed by nCounter (Nanostring) and, after normalisation of total amount of RNA molecules to the housekeeper genes, results are shown as fold change of BFCN cultures over iPSCs. Analysis of the astrocyte markers (**A**) *ALDOC*, (**B**) *SLC1A3* (EAAT1), (**C**) *NFIA* and (**D**) *S100B*. Data are derived from 10 iPSC lines and 10 BFCN differentiated samples from 1 independent differentiation for each line. Individual data are shown. Histogram bars represent mean values and error bars represent S.E.M. Graphs show *** *p* ≤ 0.001, by Kolmogorov-Smirnov test. BFCN cultures were matured for 4 weeks and immunocytochemistry for general neuronal markers and specific cholinergic neuronal markers was performed. (**E**–**G**) Representative confocal pictures of the BFCN cultures at week 4 of BFCN maturation with the nuclear marker Hoechst 33,342 in blue. (**E**) Astrocyte marker GFAP in green. (**F**) Dendritic marker MAP2 in red. (**G**) Merged picture of G and H. Scale bars = 50 μm.

**Figure 6 cells-09-02018-f006:**
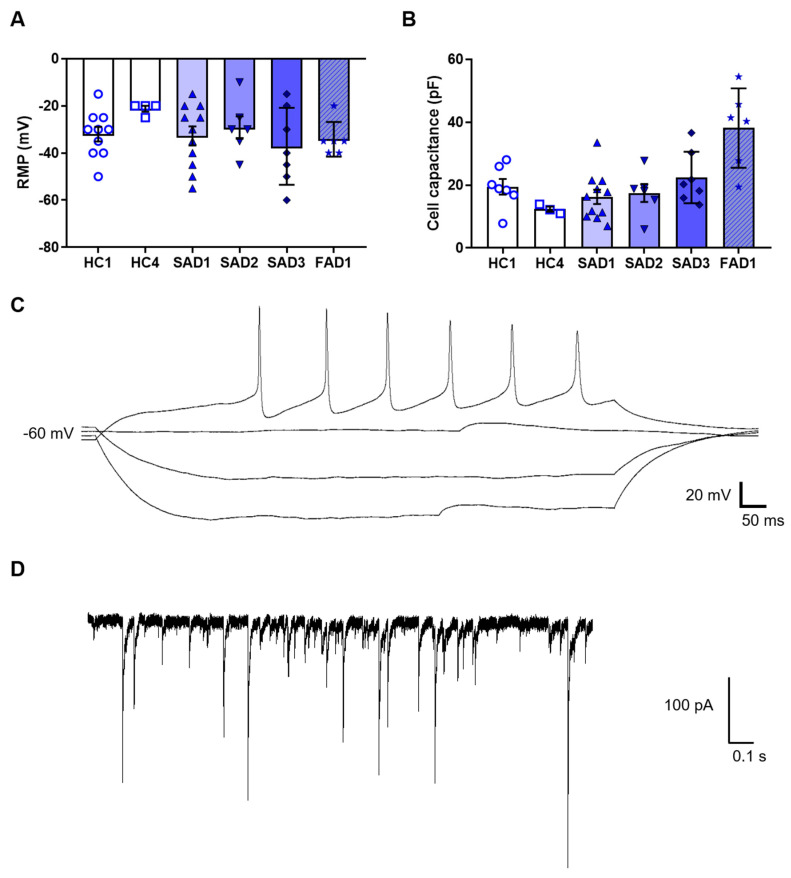
Electrophysiology data of BFCN cultures at week 4 of maturation. (**A**) Resting membrane potential and (**B**) cell capacitance. (**C**) Sample trace of a train of action potentials when membrane potential was held at −60 mV. (**D**) Sample trace of synaptic events. Data in A and B were derived from 6 BFCN differentiated samples. Individual data points representing individual cells are shown. Histogram bars represent mean values and error bars represent S.E.M.

**Table 1 cells-09-02018-t001:** List of iPSC lines used. The iPSC lines with the name used, sex and *APOE* genotype. HC, Healthy Control, SAD, Sporadic Alzheimer’s disease, FAD, Familial Alzheimer’s disease, ALS, Amyotrophic Lateral Sclerosis, FTD, Frontotemporal Dementia.

Name	iPSC Line	Sex	*APOE*	Reference
HC1	1–4	M	ε3/3	[14]
HC2	6846.2 (UOWi002-A)	F	ε2/3	[15]
HC3	RB9-8 (UOWi001-A)	F	ε2/4	[16]
HC4	MBE60-1	M	ε3/3	Appendix A
HC5	MBE68-1	F	ε3/3	[17]
SAD1	RB7-11 (UOWi006-A)	F	ε4/4	Appendix A
SAD2	8–5	M	ε4/4	[18]
SAD3	10–13	M	ε3/4	[14]
FAD1	6848.2 (UOWi003-A)	F	ε3/4	[15]
ALS/FTD1	C-10 (UOWi008-A)	F	ε3/3	Appendix A

**Table 2 cells-09-02018-t002:** List of antibodies used for immunocytochemistry.

Antibody	Raised in	Dilution	Company Cat #
Anti-β-III-tubulin	Mouse mAb	1:1000	Abcam, #ab78078
Anti-ChAT	Rabbit pAb	1:100	Abcam, #ab181023
Anti-p75	Rabbit mAb	1:50	Abcam, #ab52987
Anti-GFAP	Rabbit mAb	1:500	Millipore, #04-1062
Anti-MAP2	Mouse mAb	1:500	Millipore, #MAB3418
Anti-Rabbit IgG (H + L) Alexa Fluor 488	Goat pAb	1:1000	Life Technologies, #A11008
Anti-Mouse IgG (H + L) Alexa Fluor 633	Goat pAb	1:1000	Life Technologies, #A21050

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
