# Peer review of "A Simple Differentiation Protocol for Generation of Induced Pluripotent Stem Cell-Derived Basal Forebrain-Like Cholinergic Neurons for Alzheimer’s Disease and Frontotemporal Dementia Disease Modeling"

_cells, 2020, doi:10.3390/cells9092018_

Round 1
Reviewer 1 Report
The manuscript presented here is a novel approach to being able to define the development of Alzheimer's disease neuropathology specifically as it relates to cholinergic neurons in the basal forebrain.
As we continue to understand the disease process itself, it is apparent that precision medicine, especially as it relates to gene expression becomes more important. This study allows insight into a piece of this disease process puzzle.
Intro:
Line 43 I would prefer a citation. It is not abundantly known that AD and Picks disease effect the cholinergic system specifically. Perhaps include citation 2 there as well.
line 45 Although I agree with the sentiment of this statement, only 1 medication has been approved, at least in the US for symptom relief, memantine, and that is with limited efficacy. The focus for the last 5 years or so has been on the prevention of the further decline and/or prevention altogether. There have been robust trials focused on life style components and cognition like FINGER and MIND. I do however recognize, it is difficult to measure neuropathology in subjects who are alive, but radiographic imaging and tagging is moving in a direction to allow for that.
Did the authors think about investigating Lewy body dementia given the further interest in FT disease and that it is mentioned in the discussion?
Methods:
line 195 you wrote BFNC instead of BFCN.
Results:
no issue
Discussion
no issues.
Overall, this manuscript allows for a novel approach to how we view and model Alzheimer's disease initiation, progression and involvement of the nervous system. Further, the data presented adds to the known science and provides for a more complete picture of the diseases etiology and potentially ways in which interventions can be developed.
Author Response
- We thank the reviewer for their positive and helpful comments.
'The manuscript presented here is a novel approach to being able to define the development of Alzheimer's disease neuropathology specifically as it relates to cholinergic neurons in the basal forebrain.
As we continue to understand the disease process itself, it is apparent that precision medicine, especially as it relates to gene expression becomes more important. This study allows insight into a piece of this disease process puzzle.
Intro:
Line 43 I would prefer a citation. It is not abundantly known that AD and Picks disease effect the cholinergic system specifically. Perhaps include citation 2 there as well.'
- The manuscript has been edited as requested. Two citations have been added (line: 44).
'line 45 Although I agree with the sentiment of this statement, only 1 medication has been approved, at least in the US for symptom relief, memantine, and that is with limited efficacy. The focus for the last 5 years or so has been on the prevention of the further decline and/or prevention altogether. There have been robust trials focused on life style components and cognition like FINGER and MIND. I do however recognize, it is difficult to measure neuropathology in subjects who are alive, but radiographic imaging and tagging is moving in a direction to allow for that.'
- We agree with this comment, the focus of the statement meant to refer to drug treatments specifically. The statement has been revised to improve clarity. “Currently, the drugs that are prescribed to AD patients target symptoms and show limited efficacy [3] (line 46).
'Did the authors think about investigating Lewy body dementia given the further interest in FT disease and that it is mentioned in the discussion?'
- We agree that cholinergic neurons are highly relevant to Lewy body dementia and have specifically added this point ‘Post mortem tissue from patients with the Lewy body variant of AD (dementia with Lewy bodies) also show a loss of cholinergic neurons and reduced ChAT activity very early in the disease course [40] (line 420).
'Methods:
line 195 you wrote BFNC instead of BFCN.'
- The manuscript has been edited as requested.
'Results:
no issue
Discussion
no issues.
Overall, this manuscript allows for a novel approach to how we view and model Alzheimer's disease initiation, progression and involvement of the nervous system. Further, the data presented adds to the known science and provides for a more complete picture of the diseases etiology and potentially ways in which interventions can be developed.'
Reviewer 2 Report
In this manuscript Muñoz et al report an improved protocol for the production of iPCS-derived neuronal cells resembling basal forebrain cholinergic neurons. The application of this protocol is thought to facilitate future studies on the molecular and cellular underpinnings of familial and sporadic Alzheimer disease in vitro.
Overall the manuscript is well written, the experiments are clearly presented, and the data appear of potential interest.
A major reservation though is the question to which degree these iPSC-derived cholinergic neurons resemble their in-vivo counteracts, In fact, the data suggest that the produced neurons recapitulate cholinergic features only in part. These may be due to an immature state (see electrophysiology), incomplete cholinergic fate or both. Therefore the authors need to tone down their emphasis on the production of forebrain neurons of cholinergic identity in the title and throughout the manuscript. Alternatively, production of “basal forebrain-like cholinergic neurons” appears more adequate.
Major points
Ideally, whole transcriptomic analysis should be performed and matched with data bases on different developmental stages of human brain development and postmortem brain analysis. This would allow to broadly assign the produced neurons to developmental stages in vivo and to inform on their molecular and cellular authenticity.
Minor points
IHC experiments should be quantified.
At which passage were iPSC lines used for this study? Advanced passages may demand renewed karyotyping.
Was mycoplasma testing routinely performed?
Why were iPSC-derived neuronal cells kept without astrocytes, which are thought to boost maturation, particular with respect to the extent and quality of electrical activity? Besides, co-culture with astrocytes prevents growth in neuronal clusters as visible in Fig. 1F.
Author Response
- We thank the reviewer for their positive and helpful comments.
'In this manuscript Muñoz et al report an improved protocol for the production of iPCS-derived neuronal cells resembling basal forebrain cholinergic neurons. The application of this protocol is thought to facilitate future studies on the molecular and cellular underpinnings of familial and sporadic Alzheimer disease in vitro.
Overall the manuscript is well written, the experiments are clearly presented, and the data appear of potential interest.
A major reservation though is the question to which degree these iPSC-derived cholinergic neurons resemble their in-vivo counteracts, In fact, the data suggest that the produced neurons recapitulate cholinergic features only in part. These may be due to an immature state (see electrophysiology), incomplete cholinergic fate or both. Therefore the authors need to tone down their emphasis on the production of forebrain neurons of cholinergic identity in the title and throughout the manuscript. Alternatively, production of “basal forebrain-like cholinergic neurons” appears more adequate.'
- The manuscript has been edited as requested. We have changed the text to “basal forebrain-like cholinergic neurons” (lines: 4, 29 and 48). We agree with the comments on the immaturity of the neurons and have edited the text to clarify this point “…in this study the average resting membrane potential was -30 mV which, taken together with the high expression of some developmental markers, show the immaturity of the neurons. An increase in the maturation time over the current 4 weeks of the protocol or the presence of other cell types, such as astrocytes, would improve maturation of neurons in culture.” (lines 408-412).
'Major points
Ideally, whole transcriptomic analysis should be performed and matched with data bases on different developmental stages of human brain development and postmortem brain analysis. This would allow to broadly assign the produced neurons to developmental stages in vivo and to inform on their molecular and cellular authenticity.'
- We agree with this comment that RNAseq or similar analysis would reveal the full picture of the cellular transcriptional program, however this is not possible at this time and not within the timeframe given for revisions. Whilst we agree the data would be interesting, it is therefore beyond the scope of this current manuscript.
'Minor points
IHC experiments should be quantified.'
- We agree that the IHC experiments should be quantified. We were able to accurately quantify the ChAT+ve neurons, which is the most relevant marker for the BFCNs, as stated in the text, “Visual counting showed that 87% of the β-III-tubulin+ve cells were also ChAT+ve (Figure 2A-C)” (line: 232). We did not find GFAP+ve cells in the cultures (zero GFAP+ve cells), which was confirmed by mRNA quantification of other astrocytic markers (line 324). The growth of the cells in clusters, as seen in Figure 2, renders cell counts for some of the neurite markers inaccurate, hence their qualitative rather than quantitative description in the text. We have amended the text to clarify that the staining for P75NTR was identified in β-III-tubulin+ve cells (line 235).
'At which passage were iPSC lines used for this study? Advanced passages may demand renewed karyotyping.'
- The manuscript has been edited as requested: “Karyotyping of all the iPSC lines was performed within 20 passages before the start of the differentiation” (lines: 88-89).
'Was mycoplasma testing routinely performed?'
- We have added a statement to address this point: “iPSC cultures were tested routinely for mycoplasma” (line: 88).
'Why were iPSC-derived neuronal cells kept without astrocytes, which are thought to boost maturation, particular with respect to the extent and quality of electrical activity? Besides, co-culture with astrocytes prevents growth in neuronal clusters as visible in Fig. 1F.'
- We completely agree that co-culture with astrocytes would boost the maturity of the neurons in culture, in line with a number of studies. The text has been edited in the discussion to improve the clarity of this point: “This protocol was developed to allow for analysis of neurons, without contamination from astrocytes, as shown by the lack of GFAP staining and other astrocyte markers” (line 403). “An increase in the maturation time over the current 4 weeks of the protocol or the presence of other cell types, such as astrocytes, would improve maturation of neurons in culture [36]. Analysis of co-cultures can be the subject of future research” (line 412).